# SDNN24 Estimation from Semi-Continuous HR Measures

**DOI:** 10.3390/s21041463

**Published:** 2021-02-20

**Authors:** Davide Morelli, Alessio Rossi, Leonardo Bartoloni, Massimo Cairo, David A. Clifton

**Affiliations:** 1Huma Therapeutics Limited, London SW1P 4QP, UK; leonardo.bartoloni@huma.com (L.B.); massimo.cairo@huma.com (M.C.); 2Department of Engineering Science, Institute of Biomedical Engineering, University of Oxford, Oxford OX1 2JD, UK; david.clifton@eng.ox.ac.uk; 3Department of Computer Science, University of Pisa, 56126 Pisa, Italy; alessio.rossi2@gmail.com

**Keywords:** SDNN, HRV, HR, Logistic Regression, neural network, cardiovascular risk

## Abstract

The standard deviation of the interval between QRS complexes recorded over 24 h (SDNN24) is an important metric of cardiovascular health. Wrist-worn fitness wearable devices record heart beats 24/7 having a complete overview of users’ heart status. Due to motion artefacts affecting QRS complexes recording, and the different nature of the heart rate sensor used on wearable devices compared to ECG, traditionally used to compute SDNN24, the estimation of this important Heart Rate Variability (HRV) metric has never been performed from wearable data. We propose an innovative approach to estimate SDNN24 only exploiting the Heart Rate (HR) that is normally available on wearable fitness trackers and less affected by data noise. The standard deviation of inter-beats intervals (SDNN24) and the standard deviation of the Average inter-beats intervals (ANN) derived from the HR (obtained in a time window with defined duration, i.e., 1, 5, 10, 30 and 60 min), i.e., $ANN=\frac{60}{HR}$ (SDANN${}_{HR}$24), were calculated over 24 h. Power spectrum analysis using the Lomb-Scargle Peridogram was performed to assess frequency domain HRV parameters (Ultra Low Frequency, Very Low Frequency, Low Frequency, and High Frequency). Due to the fact that SDNN24 reflects the total power of the power of the HRV spectrum, the values estimated from HR measures (SDANN${}_{HR}$24) underestimate the real values because of the high frequencies that are missing. Subjects with low and high cardiovascular risk show different power spectra. In particular, differences are detected in Ultra Low and Very Low frequencies, while similar results are shown in Low and High frequencies. For this reason, we found that HR measures contain enough information to discriminate cardiovascular risk. Semi-continuous measures of HR throughout 24 h, as measured by most wrist-worn fitness wearable devices, should be sufficient to estimate SDNN24 and cardiovascular risk.

## 1. Introduction

The standard deviation of the inter-beats interval between QRS complexes recorded during 24 h (SDNN24) is considered the gold standard of Heart rate variability (HRV) features for cardiac health [1]. In particular, in Kleiger et al. [2] was found that people with SDNN24 values below 50 milliseconds (ms), between 50 and 100 ms and above 100 ms could be considered as unhealthy, compromised health and healthy, respectively Kleiger et al. [2]. For example, patients with SDNN24 values over 100 ms have been found to have a 5.3 times lower risk of mortality at follow-up than those with values under 50 ms [2]. The authors in Gao et al. [3] and Karcz [4] found that values of SDNN24 below 80 ms were predictive of cardiac events. Moreover, people with low HRV values are found to be associated with a high risk of a first cardiovascular event (32–45% more likely than people with normal SDNN24 values). In particular, an increase in SDNN24 of 1% results in an lower risk of heart failure of about 1% [5]. It was also found that the alteration in cardiac autonomic function is a predictor of several other individual health problems, such as diabetes [6], hypertension [7], and sleep quality [8]. Moreover, one of the last findings about SDNN24 is that changes in autonomic nervous system function that can be estimated by change in heart rate variability (HRV) have also been found to be a predictor of infection and, in particular, for the diagnosis of COVID-19 and its related symptoms [9,10,11]. As a matter of fact, a decline in HRV may be a signal of COVID-19 before common symptoms, e.g., dry cough or fever. Monitoring HRV and HR changes may thus help to evaluate the course of this virus.

The fact that SDNN24 is a strong indicator of cardiac health can be explained studying the effect of Sinoatrial Node activity on the HRV spectrum [12]. Sinoatrial Node is the natural pacemaker of the heart; by analysing its activity, information about cardiac health can be extracted [12]. The HRV spectrum is believed to be influenced mainly by the activity of the Sympathetic Nervous System, Parasympathetic Nervous System, and Sinoatrial Node. The Parasympathetic Nervous System and Sympathetic Nervous System are visible in the Low Frequency (LF) and High Frequency (HF) regions; the activity of SAN is influenced by circadian mechanisms [13] and is responsible of the characteristic 1/f shape, contributing mostly to the Ultra Low (ULF) and Very Low Frequency (VLF) ranges [14]. The LF and HF bands (affected mostly by Sympathetic Nervous System and Parasympathetic Nervous System) cover the range between 0.04 and 0.4 Hz; therefore, signals with period between 2.5 and 25 s. ULF and VLF (affected mostly by Sinoatrial Node) are affected by signals with frequencies lower than 0.04 Hz; therefore, signals with a period longer than 25 s. The HRV spectrum in the ULF and VLF behaves as 1/f; therefore, the power in the ULF region (signals with period longer than 5 min) will have more importance than the power in the VLF (signals with period between 5 min and 25 s).

SDNN24 is HRV feature that requires 24 h of continuous recording Inter-Beat Intervals, traditionally achieved using a Holter device, that makes the data collection difficult during people’s everyday life, therefore not performed routinely. Thanks to the technological advancements of recent decades, it is now possible and affordable to continuously record heart beats during 24 h via wrist-worn wearable devices equipped with heart rate sensors [15]. The low cost of these devices and their unobtrusiveness allows the larger part of the population to continuously and passively measure their heart activity. Wrist-worn wearable devices equipped with heart rate sensors have a great potential impact on the preventative health field, because is now possible to estimate the users’ health status, capturing early signs of cardiac health deterioration [16]. Due to the fact that instruments able to record inter-beats intervals, such as wrist-worn wearable devices, are more comfortable to be worn by people during daily life compared to medical devices (e.g., Holter). They could, in theory, be used to estimate SDNN24. However, inter-beats intervals recorded from these devices suffer from high amount of noise and motion artifact, that propagate to HRV features [17,18]. Furthermore, wrist-worn devices normally only report heart rate data (HR), as these have been proven to be more reliable than inter-beats intervals data which can only be estimated from these devices in both resting and during physical activity with a small error [16,17,18,19,20,21], that seldom report the inter-beats intervals data needed to compute SDNN24.

Abnormal SAN activity affects the HRV spectrum [22]. Since SAN activity generates frequencies with a 1/f profile [23], changes in HRV will be more visible in the ULF and VLF frequency ranges of HRV [14,24]. HR (the reciprocal of the average inter-beats intervals duration over 1 min) can be considered a low pass filter on the inter-beats intervals signal. A low pass filter that lets frequencies lower than 1 min pass will not affect ULF and VLF that are the main contributors of SDNN24. Therefore, it should, in theory, be possible to estimate SDNN24 using measures of HR, collected over 24 h. If our hypothesis can be verified, this study could show that it is possible to estimate SDNN24 from semi-continuous HR measures and continuous inter-beats intervals data are not strictly necessary. This makes the SDNN24 estimation particularly resilient to noisy data (i.e., unevenly sampled data with huge quantity of missing data induced by motion artefacts) a condition normally found with wrist-worn wearable devices. Furthermore, the second aim of this study is to assess that HR data are sensible to discriminate healthy and unhealthy people such SDNN24 do. If this supposition will be verified, the HR data—that are less sensible to missing values [17]—could be used as a predictor of cardiovascular disease and the lack of usable inter-beats intervals data (more difficult to be reliably measured by wrist-worn fitness wearable devices with heart rate sensors) would not necessarily prevent continuous cardiac health estimation.

## 2. Methods

### 2.1. Dataset and Feature Engineering

In this paper, we used two PhysioNet datasets, providing about 23 h of electrocardiograms (ECG) data of 105 healthy and unhealthy subjects:nsr2db: Normal Sinus Rhythm RR Interval Database PhysioNet dataset [25]. This dataset contains beat annotations of 54 normal sinus rhythm subjects (30 men: 28–76 years; 24 women: 58–73 years) extracted from 23 h long ECG.chfdb: Congestive Heart Failure RR Interval Database PhysioNet dataset [26]. This dataset includes beat annotation files for 29 long-term ECG recordings of subjects aged 34 to 79, with congestive heart failure (New York Heart Association classes I, II, and III). Subjects included 8 men and 2 women; gender is not known for the remaining 21 subjects.mmash: Multilevel Monitoring of Activity and Sleep in Healthy people (MMASH) dataset [27,28] provides 24 h of continuous beat-to-beat heart data, triaxial accelerometer data, sleep quality, physical activity and psychological characteristics (i.e., anxiety status, stress events and emotions) for 22 healthy participants.

These data are used to simulate the HR data extracted from wrist-worn wearable devices equipped with heart rate sensors. Typically, these devices still only measure the average heart rate in a semi-continuous way during the day [29] with a different granularity in accordance with battery saving policies arbitrarily decided by the device manufacturers. In Section 2.3, we estimate the error of HR values recorded by using PPG devices.

### 2.2. Data Preprocessing

The python hrv-analysis library (https://pypi.org/project/hrv-analysis, accesed on: 1 September 2020) is used to remove outliers and ectopic beats from signal in each long-term ECG timeseries, as showed in Rossi et al. [30]. In total, 2.46 ± 4.29% of inter-beats intervals in the dataset were ectopic beats (i.e., disturbance of the cardiac rhythm frequently related to the electrical conduction system of the heart) or missing values induced by motion artefacts. These missing values were reconstructed via quadratic interpolation applied on the time domain (i.e., the heartbeats timestamp) as suggested by Morelli et al. [17].

### 2.3. PPG Error Estimation

PPG devices are highly affected by motion artefacts that could induce an error in HR estimation. However, due to the fact that HR provided by these devices is the mean of HR values during a time window arbitrarily defined by device brand, it is equivalent to applying a low pass filter on instant HR data that permit us to remove most of the noise in the time series. In order to assess this error, we compare the average of the instant HR in different time window length (i.e., 1, 5, 10, 30 and 60 min) between ECG (Polar H7 heart rate monitor—Polar Electro Inc., Bethpage, NY, USA) and PPG (BioBeam—BioBeats group Ltd., London, UK, www.biobeats.com) data from MMASH dataset. Unfortunately, due to technical issues during data recording, we have HR data for both the devices only for a small, but still significant, parts of the day (only a few hours per participant; not enough to be used as the main dataset for the paper, but enough to characterise PPG sensor noise vs. ECG ground truth). The mean and standard deviation of HR estimation error for different window lengths (i.e., 1, 5, 10, 30 and 60 min) are provided in Table 1. It can be seen that the higher the time window, the lower the HR bias variability. This result is due to the fact calculating the average heart rate is equivalent to applying a low pass filter on instant HR data, therefore, limiting the influence of outliers caused by motion artefacts on the average HR.

### 2.4. SDNN24

#### 2.4.1. Time Domain Analysis

From each user, the standard deviation of inter-beats intervals (NN) over 24 h (SDNN24) were computed in four different ways:SDNN24: The standard deviation of NN intervals recorded during 24 h (Equation (Equation 1)).
(1)$$\mathrm{SDNN}24=\sqrt{\frac{\sum_{i=1}^{N}{\left({\mathrm{NN}}_{i}-\bar{\mathrm{NN}}\right)}^{2}}{N}}$$
where ${\mathrm{NN}}_{i}$ and $\bar{\mathrm{NN}}$ refer to each NN-interval value and the mean of 24 h NN-intervals, respectively. *N* is the number of NN-intervals recorded during 24 h. From a digital signal processing perspective, SDNN24, being the standard deviation of the 24 h long signal, is the power of the whole 24 h spectrum. As discussed in the introduction, we expect ULF and VLF to be the main contributors to the spectrum; therefore, we expect SDNN24 to correlate with VLF and ULF. This is the definition of SDNN24, and the ground truth we will try to estimate. To compute SDNN24 is necessary to have the continuous stream of inter-beats intervals signal, typically not available from wearable devices.SDNNi24: The mean of the standard deviations of the NN intervals calculated on segments with defined duration over 24 h (Equation (Equation 2)).
(2)$$\mathrm{SDNNi}24=\frac{\sum_{j=1}^{N_{\mathrm{segments}}}\sqrt{\frac{\sum_{i=1}^{n_{j}}{\left({\mathrm{NN}}_{i}-\bar{{\mathrm{NN}}_{j}}\right)}^{2}}{n_{j}}}}{N_{\mathrm{segments}}}$$
where $N_{\mathrm{segments}}$ and $n_{j}$ reflect the number of segments and the number of NN-intervals in each segment, respectively. From a digital signal processing perspective, SDNNi24, being the standard deviation of short term measurements (usually 1 to 5 min), represents the average power of the short term spectrum. It will not be able to measure ULF and VLF (that comes from signals of periods longer than 5 min). To compute SDNN24 is necessary in order to have the inter-beats intervals data of each segment, typically not available from wearable devices.SDANN24: The standard deviation of the means of NN intervals calculated at segments of a defined duration over 24 h (Equation (Equation 3)).
(3)$$\mathrm{SDANN}24=\sqrt{\frac{\sum_{j=1}^{N_{\mathrm{segments}}}{\left(\bar{{\mathrm{NN}}_{j}}-\frac{\sum_{j=1}^{N_{\mathrm{segments}}}\bar{{\mathrm{NN}}_{j}}}{N_{\mathrm{segments}}}\right)}^{2}}{N_{\mathrm{segments}}}}$$
where $\bar{{\mathrm{NN}}_{j}}$ refers to the mean of the NN-intervals in a segment. $N_{\mathrm{segments}}$ is the number of NN-intervals time windows recorded over 24 h. From a digital signal processing perspective SDANN24 can be considered similar to SDNN24 applied on a inter-beats intervals dataset after a low pass filter that dampened signals with a shorter period than the duration of the measured segments. To compute SDNN24, it is necessary to have the inter-beats intervals data of each segment, typically not available from wearable devices.SDANN${}_{\mathrm{HR}}$24: The standard deviation of the Average NN intervals (ANN) derived from the HR, i.e., $\mathrm{ANN}=\frac{60}{\mathrm{HR}}$, calculated on segments with defined duration over 24 h (Equation (Equation 4)).
(4)$${\mathrm{SDANN}}_{\mathrm{HR}}24=\sqrt{\frac{\sum_{j=1}^{N_{\mathrm{segments}}}{\left(\frac{60}{{\mathrm{HR}}_{j}+\epsilon_{j}}-\frac{\sum_{j=1}^{N_{\mathrm{segments}}}\frac{60}{{\mathrm{HR}}_{j}+\epsilon_{j}}}{N_{\mathrm{segments}}}\right)}^{2}}{N_{\mathrm{segments}}}}$$
where HR is computed as 60/$\bar{{\mathrm{NN}}_{\mathrm{segments}}}$. SDANN${}_{\mathrm{HR}}$24 can be computed from data commonly collected by wrist-worn fitness wearable devices. In order to simulate error induced by wrist-worn devices, we randomly add a Gaussian error ϵ in each HR obtained during each NN-intervals time window. The bias for each time window’s length is provided in Table 1.

#### 2.4.2. Frequency Domain Analysis for SDNN24

The power spectrum $S_{xx}\left(f\right)$ of a time series x(t) describes the distribution of power into frequency components (ω) composing that signal. Due to the fact that the inter-beats intervals are not uniformly distributed (unevenly sampled time-series) the Power Spectrum Density (PSD) are computed using the Lomb–Scargle Periodogram [31,32] instead of the Fourier Transformation that requires uniformly sampled data. The PSD can be used to compute the variance (net power) of a process by integrating over frequencies as showed in Equation (Equation 5).
(5)$$\mathrm{VAR}\left(x\left(t\right)\right)=\frac{1}{\pi }\int_{0}^{\infty }S_{xx}\left(\omega \right)d\omega$$

Consequently, it is possible to estimate SDNN24 by PSD derived from a 24-h time-series by the square of its net power (VAR(x(t))). Due to the fact that SDANN${}_{\mathrm{HR}}$24 reflects frequencies after a low pass filter that corresponds to the segments length, missing frequencies induce an underestimation of SDNN24. For example, if the 60/HR time-series provide values over 5 min, it is possible to assess only frequency lower than 0.0033. For these reasons we computed SDNN24 in 2 other ways by adding the estimated variance of the missing frequency on the variance of 60/HR time-series with a defined segments length over the 24 h. To this aim, we computed apriori PSD on nsr2db and chfdb and tested the validity of this approach on mmash dataset.

We evaluate two ways of correcting the bias of discarding high frequencies when using any approach based on HR measures:SDANN${}_{\mathrm{HR}}$24${}_{\mathrm{adjMean}}$: The root square of the sum between NN intervals (ANN) variance derived from the average HR, i.e., ANN = $\frac{60}{{\mathrm{HR}}_{segment}}+\epsilon$ (ϵ is a random Gaussian bias of a specific time window length), calculated on segments with defined duration over 24 h (ANN${}_{\mathrm{HR}}$24) and the mean of apriori missing frequency variance (Equation (Equation 6)).
(6)$$\;{\mathrm{SDANN}}_{\mathrm{HR}}24_{\mathrm{adjMean}}=\sqrt{\mathrm{VAR}\left({\mathrm{ANN}}_{\mathrm{HR}}24\right)+\frac{1}{\pi }\int_{freq}^{\infty }S_{xx}\left(\omega \right)d\omega }$$
where freq refers to the length of the segment in $\frac{60}{\mathrm{HR}}$ time-series (e.g., for 1, 5 and 10 min segments the freq are equal to 0.016, 0.0033 and 0.0016, respectively).With this approach, we attempt to remove the underestimation of SDNN24 by adding the portion of spectrum lost by using HR measures instead of inter-beats intervals data, simply adding the average power of the HRV spectrum above freq to the measured variance. The corrective factor is fixed for all subjects.SDANN${}_{\mathrm{HR}}$24${}_{\mathrm{adjW}}$: The root square of the total power predicted in accordance with apriori PSD (Equation (Equation 7)).
(7)$${\mathrm{SDANN}}_{\mathrm{HR}}24_{\mathrm{adjW}}=\sqrt{\frac{1}{\pi }\mathrm{VAR}\left({\mathrm{ANN}}_{\mathrm{HR}}24\right)+\frac{1}{\pi }\frac{\int_{0}^{freq}S_{xx}\left(\omega \right)d\omega_{ANN}*\int_{freq}^{\infty }S_{xx}\left(\omega \right)d\omega }{\int_{0}^{freq}S_{xx}\left(\omega \right)d\omega }}$$
where freq refers to the length of the segment in $\frac{60}{\mathrm{HR}}$ time-series (e.g., for 1, 5 and 10 min segments the freq are equal to 0.016, 0.0033 and 0.0016, respectively).With this approach we correct the underestimation by assuming that the missing high frequencies perfectly correlate with the measured low frequencies.

#### 2.4.3. HR Circadian Rhythm

A multiple component cosinor model (Equation (Equation 8)) was fitted on HR values recorded over 24 h in order to assess the HR circadian rhythm [33].
(8)$${\mathrm{HR}}_{t}=M+A\times cos\left(2\pi \times \left(\frac{t}{\mathrm{freq}}+\phi \right)\right)+\epsilon_{t}$$
where *M* is the MESOR (Midline Estimating Statistic Of Rhythm, a rhythm-adjusted mean), *A* is the amplitude (a measure of half the extent of predictable variation within a cycle), ϕ is the acrophase (time of the day when the high HR values recurs in each cycle), *t* is the period (duration of one cycle), freq is the fixed length of the cycle (i.e., 24 h) and $\epsilon_{t}$ is the error term. The circadian parameters were then used as features for Machine Learning models, described in the remainder of this manuscript.

### 2.5. Validation

The statistical differences between SDNN24 (gold standard) and SDNNi24, SDANN24, and SDANN${}_{HR}$24 were assessed by paired *t*-test, while their relationship was evaluated by Pearson correlation coefficient. All the Physionet datasets were involved in this analysis.

The Power Spetrum Density (PSD) of the 24 h RR time-series was estimated using the Lomb-scargle Periodogram [31,32]. APriori PSD was created on chfdb and nsr2db and was used to estimate SDANN${}_{\mathrm{HR}}$24${}_{\mathrm{adjMean}}$ and SDANN${}_{\mathrm{HR}}$24${}_{\mathrm{adjW}}$ on the mmash dataset. This approach was used in order to avoid any overfitting problems caused by assessing missing frequency on the same data that were used to predict SDNN24.

Paired *t*-test, Pearson correlation coefficient and Root Mean Squared Error (RMSE) were assessed to evaluate statistical difference, relationship and error between gold standard and estimated features. All the analysis was conducted on 1, 5, 10, 30 and 60 min segments of time in order to assess the maximal time window that provides reliable results.

### 2.6. Healthy vs. Unhealthy Subjects

#### 2.6.1. Statistical Analysis

People in the nsr2db and chfdb dataset with moderate and severe congestive heart failure (New York Heart Association class 2–3–4) were labelled as high cardiovascular risk, while people with class 0–1 were labelled as low risk. HRV features (i.e., ULF, VLF, LF and HF), SDNN24 and SDNN${}_{\mathrm{HR}}$24 (computed on 5 min time window differences) between people with low and high cardiovascular risk were assessed by unpaired *t*-test. The T-score derived from *t*-test analysis was used to assess the magnitude of the difference between healthy vs. unhealthy subjects.

#### 2.6.2. Machine Learning Approach

In order to discriminate between healthy and unhealthy subjects, grouped as shown in Section 2.6.1, the following Machine Learning (ML) models were evaluated.

LR: Logistic Regression was performed using: only SDNN24 (${\mathrm{LR}}_{\mathrm{SDNN}}24$); only SDNN${}_{\mathrm{HR}}$24 ($LR_{\mathrm{SDNN}}\mathrm{HR}24$); all of the HR features as predictors, i.e., SDNN${}_{\mathrm{HR}}$24, MESOR and Amplitude ($LR_{\mathrm{HR}}$);RF${}_{\mathrm{HR}}$: Random Forest Classifiers (RF) were also performed using all the HR features as predictors;NN: Fully connected feed forward Neural Network, using all the HR features as predictors. We used Keras with the TensorFlow backend by using Python 3.8 programming language. We trained our neural networks on the Azure cloud, using bayesian sampling. The only explored topology was fully connected, with a single hidden layer, *leaky relu* activation function for the neurons of the hidden layer, a single output neuron with *sigmoid* activation function, and a dropout layer after the hidden layer. The tuned hyper-parameters were:
The number of neurons in the hidden layer (between 1 and 8);Alpha value for the *leaky relu* activation function of the neurons in the hidden layer (between 0.0 and 1.0);The dropout rate (between 0% and 99%);The batch size (between 1 and 32). The training set was split in train and validation using the *train_test_split* function from the *sklearn* python package, using a 80-20% split, ensuring stratification on the predicted class. The validation data were not used during hyper-parameter tuning. A total of 400 combinations of hyper-parameters were tested.

In order to assess the validity of the classifiers we compared our predictive models with two baselines. Baseline B1 randomly assigned a class to an example by respecting the distribution of classes, while Baseline B2 always assigned the majority class.

The models were trained on 70% of the dataset and tested on the remains 30%, and the goodness of the predictions were assessed using Precision, Recall and F1-score.

## 3. Results

### 3.1. SDNN24 Estimation

#### 3.1.1. Time Domain Analysis

Descriptive statistics of all the SDNN24 estimations computed on time domain are provided in Table 2. Statistical differences (*p* < 0.001) were detected for each estimated feature on each segment length. The longer the segment length, the higher the difference among SDNN24, SDANN24 and SDANN${}_{\mathrm{HR}}$24. Strong positive correlations were detected between SDNN24 and SDANN24 (r = [0.95–0.96]) and SDANN${}_{\mathrm{HR}}$24 (r = [0.93–0.94]) in all of the segment lengths. Differently, the mean of the standard deviations of the NN intervals calculated on defined segments during 24 h shows an opposite trend. The higher the segment length the lower the difference between SDNN24 and SDNN${}_{i}$24. Moreover, a moderate relationship was found between SDNN24 and SDNN${}_{i}$24 (r = [0.67–0.71]).

#### 3.1.2. Frequency Domain Analysis

Figure 1 provides the power spectrum of the 24 h RR-intervals time series on users in nsr2db and chfdb datasets. SDNN24 measures the total power of the spectrum of the analysed data, therefore, including both ultra low (ULF), very low (VLF), low (LF), and high frequencies (HF), as well as the frequencies above HF [1]. When analysing human HRV, ULF are defined as the fluctuations with a period between 5 min and 24 h, therefore, capturing mainly circadian effects; VLF measures oscillations between 25 s and 5 min; LF measures oscillations between 7 and 25 s, is known to be influenced by the activity of both the Parasympathetic Nervous System and the Sympathetic Nervous System; HF measures oscillations between 7 and 2 s, and is known to be influenced by Sympathetic Nervous System and the respiratory rate. The strong linear relationship (r = 0.98, *p*-value < 0.001) detected between ULF and Total Power (i.e., integration of the spectral components) suggests that it is possible to accurately predict Total Power and consequently SDNN24 from ULF.

Lower values obtained from SDANN24 and SDANN${}_{\mathrm{HR}}$24 compared to SDNN24 (see Table 2) is due to the fact that only a subset of the spectrum are measured because of the segment length. For example, in 5 min segments, it is possible to assess frequencies lower than 0.0033 (i.e., ULF), while higher frequencies are missing. Even if the low frequency reflects a huge part of the total power (Table 3), the missing frequency does not permit it to accurately estimate SDNN24, underestimating it.

Table 4 shows that SDANN${}_{\mathrm{HR}}$24${}_{\mathrm{adjW}}$ is able to accurately estimate SDNN24 with all the segments length with a bias from 2.18 ± 21.06 ms (RMSE = 21.17 ms) to 15.23 ± 26.86 ms (RMSE = 28.40 ms) for 1 min and 60 min segments, respectively. The strong correlation (r > 0.97) found between low frequencies and total power in all of the segment lengths allows us to accurately estimate the total power and consequently SDNN24. Differently, only 1 min segments show reliable results for SDNN${}_{\mathrm{HR}}$24 (bias = 9.54 ± 23.04 ms, RMSE = 24.94 ms, *p*-value = 0.07), but any segments show similar results for SDANN${}_{\mathrm{HR}}$24${}_{\mathrm{adjMean}}$.

### 3.2. Healthy vs. Unhealthy Subjects

Figure 2 shows that the differences between people with low and high risk of cardiovascular disease in PSD are found on ULF (*p*-value < 0.001) and VLF (*p*-value < 0.001). In accordance with the fact that low frequency (ULF and VLF) has a greater effect on people’s cardiovascular health than other frequencies (LF and HF), and Table 5 shows that SDNN${}_{\mathrm{HR}}$24 provides more differences (t-score) between people with low and high cardiovascular risk compared to SDNN24. Similar results are detected from others segment lengths. Hence, SDNN${}_{\mathrm{HR}}$24 is more sensitive to discriminating between healthy and not-healthy people, due to the fact that it assesses only the low frequencies—which is more sensitive to cardiovascular risk level, as shown in Figure 2—instead of the total power such as SDNN24.

The ML models demonstrate that HR parameters permit to accurately predict SDNN24. In particular, the forward stepwise linear regression shows that it is possible to predict SDNN24 by using HR circadian rhythm parameters and SDANN${}_{\mathrm{HR}}$24 as shown in Equation (Equation 9). A very low error is detected between SDNN24, predicted and observed showing a difference of about 0.22 ± 11.47 (RMSE = 53.81 and $r^{2}$ = 0.97).
(9)$$\mathrm{SDNN}24=47.248+951.590\times {\mathrm{SDANN}}_{HR}24-0.347\times \mathrm{MESOR}$$

Moreover, HR is found to be also slightly more informative than SDNN24 to discriminate healthy and unhealthy people improving the prediction performance of about 9%. In particular, the higher prediction performance is detected by ${\mathrm{LR}}_{\mathrm{HR}}$ (Table 6). Equation (Equation 10) shows the LR function that best discriminates subjects with high risk of cardiovascular diseases. As expected, the fully connected neural network (NN) also reaches high predictive capabilities. However, since the predictive performances of $LR_{\mathrm{HR}}$ and NN are comparable, ${\mathrm{LR}}_{\mathrm{HR}}$ should be preferred over NN, due to its simplicity and interpretability.
(10)$$f\left(x\right)=\frac{1}{1+e^{0.121+0.283*{\mathrm{SDANN}}_{\mathrm{HR}}24-0.293*\mathrm{MESOR}+0.225*\mathrm{Amplitude}}}$$

## 4. Discussion

This study has shown that SDNN24 may possibly be estimated from HR data, without the need to have 24-hour inter-beat intervals data available. Continuous or semi-continuous HR measures are nowadays affordable via wearable devices equipped with a PPG sensor, such as fitness trackers. Estimating HRV features from wearable devices is known to be problematic because of noise induced by motion artefacts that affect the frequencies normally used to assess the activity of the Autonomic Nervous System, such as the Root Mean Square of Successive Differences (RMSSD) and the Sympathovagal Imbalance (SVI) [17,18]. However, HR is calculated by taking the average of the duration of the inter-beat intervals’ time-series that is equivalent to applying a low pass filter that permits one to filter out most of the noise.

In theory, to truly estimate SDNN24, we need precise information from the whole power spectrum, i.e., from ULF to HF and above. However, analysing the normative values of the human HRV power spectrum (see Table 3 and Figure 1), we notice that the power of ULV is larger than the power of VLF by several orders of magnitude. In turn, the power of VLF is larger than LF which, in turn, is larger than the power of HF. The contribution of HF and LF to SDNN24 is, therefore, a residual fraction of SDNN24, compared to the importance of ULF and VLF [34]. As discussed in the introduction, the activity of the Sinoatrial Node is indicative of cardiac health [12] and responsible for the characteristic 1/f shape of the HRV spectrum [14]. Therefore ULF and VLF will yield more information about Sinoatrial Node activity than LF and HF.

Because semi-continuous HR measures are equivalent to measuring the heart activity after a low pass filter, it is a stable proxy for low HRV frequencies. Due to the fact that SDNN24 reflects the total power of the spectrum, the values estimated by splitting into segments the inter-beats intervals timeseries (i.e., SDNN*i*24, SDANN24 and SDANN${}_{\mathrm{HR}}$24) underestimate the real values because of the missing high frequencies (Table 2 and Table 3). However, the higher percentage of the total power comes from ultra and very low frequencies (Table 3), it is, therefore, possible to estimate SDNN24 with a bias. Table 2 shows that the length of the segments affects the magnitude of the bias; the shorter the segments where HR are obtained, the lower the bias between SDNN24 and estimated values.

By adding the expected power spectrum of the missing frequencies to the variance estimated via HR measures (i.e., SDANN${}_{\mathrm{HR}}$24${}_{\mathrm{adjMean}}$ and SDANN${}_{\mathrm{HR}}$24${}_{\mathrm{adjW}}$), it is possible to obtain accurate estimates of SDNN24. In particular, Table 4 shows that no statistical differences were detected between SDNN24 and SDANN${}_{\mathrm{HR}}$24${}_{\mathrm{adjW}}$ in all the segments’ lengths, while no statistical difference was only detected with 1 min segments for SDANN${}_{\mathrm{HR}}$24${}_{\mathrm{adjMean}}$. The strong linear relationship detected between the power of the low frequencies and the total power (r > 0.98) permits one to accurately estimate the power of the total spectrum by assessing only a small part of that (SDANN${}_{\mathrm{HR}}$24${}_{\mathrm{adjW}}$). Differently, adding a constant power of the missing frequency observed in apriori PSD is permitted at the variance of ANN computed with different time-window length (SDANN${}_{\mathrm{HR}}$24${}_{\mathrm{adjMean}}$) is not a good proxy for SDNN24 because the power of the missing frequencies are not weighted in accordance with the magnitude of the low frequencies observed. Obviously, the shorter the time window, the lower the bias between SDNN24 and SDANN${}_{\mathrm{HR}}$24${}_{\mathrm{adjMean}}$ due to less missing frequencies.

Figure 2 shows that people with low and high cardiovascular risk show a different power spectrum. In particular, higher ULF and VLF were detected for people with low cardiovascular risk compared to high risk ones, while similar results are shown for LF and HF. For this reason, SDNN${}_{\mathrm{HR}}$24, which reflect ultra and very low frequencies, showed higher difference between healthy and unhealthy people (Table 5) for all the segments length. This results is corroborated from the fact that Machine Learning models were able to classify cardiovascular risk from HR measures (Table 6). These results indicate that semi-continuous HR measures contain enough information to assess cardiac health.

It should be noted that one of the most important limitations of this study is that our approach is validated on simulated HR data derived from a wrist-worn fitness tracker by using ECG data that should not be affected by motion artefacts. However, in order to better simulate data from these low cost devices, we introduce an error of estimated from real wearable devices in accordance with the bias found in MMASH dataset as showed in Table 1. However, future works are needed in order to validate our approach on HR data recordings from wrist worn devices.

## 5. Conclusions

HR data permit us to accurately estimate SDNN24 and it is found to be also slightly more informative to discriminate healthy and unhealthy people. As a matter of fact, the information contained in HR measures over 24 h (ultra and very low frequencies), should be sufficient to estimate SDNN24 and the people health status from wrist-warn fitness wearable devices that provide only fragmentary HR data throughout the day. This result makes the SDNN24 estimation particularly resilient to noisy data and consequently could be easily and accurately estimated with wrist-worn wearable devices. This result indicates that HR fitness trackers have the potential to implement the continuous monitoring of cardiovascular health from passively collected data, that could enable targeted interventions at early signs of deterioration of Sinoatrial Node activity.

## Figures and Tables

**Figure 1 sensors-21-01463-f001:**
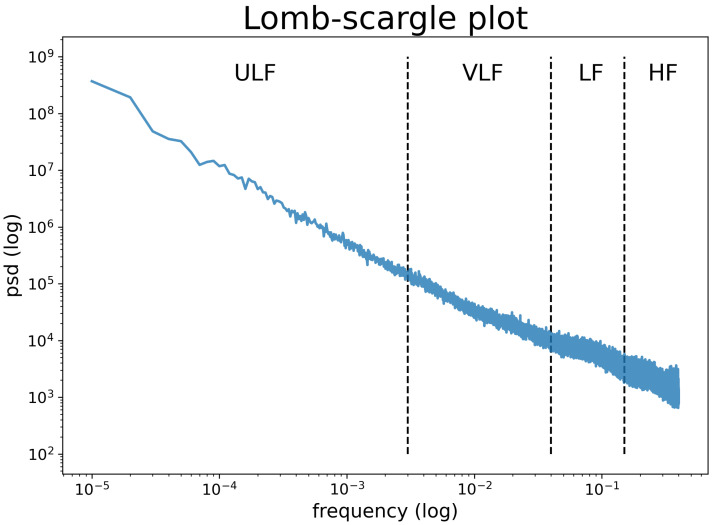
Power spectrum density plot obtained by Lomb-Scargle Periodogram.

**Figure 2 sensors-21-01463-f002:**
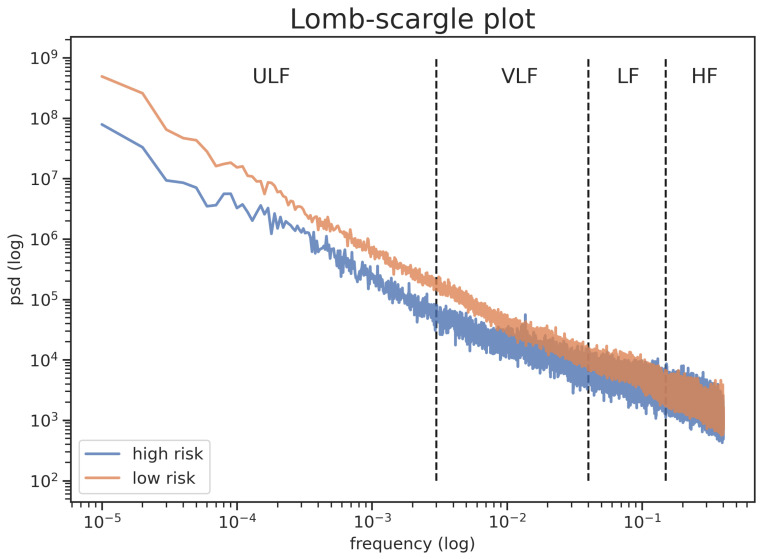
Power spectrum density plot obtained by Lomb-Scargle Periodogram in both low and high cardiovascular risk subjects.

**Table 1 sensors-21-01463-t001:** HR estimation error. The estimation errors are expressed as mean ± standard deviation.

Time Window	Error
1 min	0.03 ± 5.91
5 min	0.33 ± 5.09
10 min	−0.16 ± 4.71
30 min	−0.60 ± 4.01
60 min	−0.35 ± 3.95

**Table 2 sensors-21-01463-t002:** Descriptive and statistics analysis of SDNN24 estimations. All the values are expressed in milliseconds (ms).

Segment	SDNN24	SDNNi24	SDANN24	SDANN${}_{HR}$24
1 min	133.73 ± 45.98	56.28 ± 18.51 *	119.38 ± 49.51 *	120.96 ± 48.91 *
5 min		59.47 ± 21.52 *	117.33 ± 48.03 *	118.62 ± 47.00 *
10 min		60.77 ± 23.02 *	116.72 ± 47.37 *	117.94 ± 46.30 *
30 min		63.11 ± 26.26 *	115.75 ± 46.69 *	117.12 ± 45.96 *
60 min		64.61 ± 28.52 *	114.87 ± 46.23 *	116.29 ± 45.85 *

* *p*-value < 0.01.

**Table 3 sensors-21-01463-t003:** Descriptive of power spectrum analysis expressed in ms${}^{2}$ in all the segments length. In this table, the power of frequencies that are lower and higher than the maximal frequency that can be evaluated because of the segment length are reported. The results are reported as the mean ± standard deviation.

Segment	Lower Frequency	Higher Frequency	Total Power
1 min max frequency = 1.67$\times 10^{-2}$	1.17$\times 10^{4}$± 8.68$\times 10^{3}$ (20.44%)	4.57$\times 10^{4}$± 2.86$\times 10^{4}$ (79.56%)	5.74$\times 10^{4}$± 3.66$\times 10^{4}$
5 min max frequency = 3.33$\times 10^{-3}$ Hz	1.12$\times 10^{4}$± 8.47$\times 10^{3}$ (19.41%)	4.62$\times 10^{4}$± 2.86$\times 10^{4}$ (80.59%)	
10 min max frequency = 1.67$\times 10^{-3}$ Hz	1.09$\times 10^{4}$± 8.35$\times 10^{3}$ (18.88%)	4.66$\times 10^{4}$± 2.87$\times 10^{4}$ (81.22%)	
30 min max frequency = 5.55$\times 10^{-4}$ Hz	1.03$\times 10^{4}$± 8.09$\times 10^{3}$ (17.85%)	4.72$\times 10^{4}$± 2.90$\times 10^{4}$ (82.25%)	
60 min max frequency = 2.78$\times 10^{-4}$ Hz	9.77$\times 10^{3}$± 7.90$\times 10^{3}$ (17.01%)	4.77$\times 10^{4}$± 2.92$\times 10^{4}$ (82.98%)	

**Table 4 sensors-21-01463-t004:** Descriptive and statistics analysis of SDNN24 and SDNN24 derived from HR data on mmash dataset. All the values are expressed in milliseconds (ms).

Segment	SDNN24	SDNN${}_{\mathrm{HR}}$24	SDANN${}_{\mathrm{HR}}$24${}_{\mathrm{adjMean}}$	SDANN${}_{\mathrm{HR}}$24${}_{\mathrm{adjW}}$
1 min	173.53 ± 25.76	164.00 ± 32.52	213.63 ± 20.81 *	175.65 ± 25.65
5 min		153.02 ± 32.54 *	209.23 ± 19.79 *	165.64 ± 31.59
10 min		149.84 ± 31.00 *	198.92 ± 23.90 *	161.51 ± 30.96
30 min		146.02 ± 32.47 *	197.60 ± 24.55 *	164.21 ± 35.98
60 min		142.11 ± 35.02 *	198.38 ± 31.10 *	155.23 ± 40.00

* *p*-value < 0.01.

**Table 5 sensors-21-01463-t005:** Difference between people with low and high cardiovascular risk in all the segments length.

Segment	Features	High Risk	Low Risk	t-score
—	SDNN24 (ms)	86.54 ± 43.29	142.14 ± 31.05	6.66 *
1 min	SDNN${}_{\mathrm{HR}}$24 (ms)	67.86 ± 37.23	132.61 ± 30.79	8.28 *
5 min	SDNN${}_{\mathrm{HR}}$24 (ms)	64.46 ± 37.23	128.31 ± 30.63	8.27 *
10 min	SDNN${}_{\mathrm{HR}}$24 (ms)	62.55 ± 36.67	126.02 ± 30.52	8.29 *
30 min	SDNN${}_{\mathrm{HR}}$24 (ms)	58.37 ± 35.45	121.96 ± 30.44	8.44 *
60 min	SDNN${}_{\mathrm{HR}}$24 (ms)	55.75 ± 34.60	118.81 ± 30.26	8.49 *

* *p*-value < 0.01.

**Table 6 sensors-21-01463-t006:** Model performance.

Model	Class	Precision	Recall	F1-score
${\mathrm{LR}}_{\mathrm{SDNN}}24$	Low	0.84	1.00	0.91
	High	1.00	0.50	0.67
${\mathrm{LR}}_{\mathrm{SDNNi}}24$	Low	0.76	1.00	0.86
	High	1.00	0.17	0.29
${\mathrm{LR}}_{\mathrm{SDANN}}24$	Low	0.80	1.00	0.89
	High	1.00	0.33	0.50
${\mathrm{LR}}_{\mathrm{SDANN}}\mathrm{HR}24$	Low	0.80	1.00	0.89
	High	1.00	0.33	0.50
${\mathrm{LR}}_{\mathrm{HR}}$*	Low	0.94	1.00	0.97
	High	1.00	0.83	0.91
${\mathrm{RF}}_{\mathrm{HR}}$	Low	0.80	1.00	0.89
	High	1.00	0.33	0.50
NN	Low	0.94	1.00	0.97
	High	0.80	1.00	0.89
B1	Low	0.72	0.81	0.76
	High	0.25	0.17	0.20
B2	Low	0.73	1.00	0.84
	High	0.00	0.00	0.00

* refers to the model with high performance.

## Data Availability

Publicly available datasets were analyzed in this study.

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
