# Peer review of "SDNN24 Estimation from Semi-Continuous HR Measures"

_sensors, 2021, doi:10.3390/s21041463_

Round 1

Reviewer 1 Report

The study shows that SDNN24 can be reliably estimated from HR data, instead of a 24 hours long stream of IBI data. Interesting approach on how to improve underestimation of the PSD (due to the missing frequencies as a consequence of using Lomb–Scargle Periodogram) by adding the estimated variance of  the missing frequency, thus obtaining an accurate estimate of SDNN24.

Minor comments:

* The close parenthesis is missing in the equation 8.

* Line 171:

... the MESOR (Midline Statistic Of Rhythm, ...

It should be:

... the MESOR (Midline Estimating Statistic Of Rhythm, ...

* Table 5, Abbreviations: Prec, Rec, and F1,  were not defined earlier.

In Table 5 should be:

Precision, Recall,  and F1-score.

Author Response

Reviewer comments: The study shows that SDNN24 can be reliably estimated from HR data, instead of a 24 hours long stream of IBI data. Interesting approach on how to improve underestimation of the PSD (due to the missing frequencies as a consequence of using Lomb–Scargle Periodogram) by adding the estimated variance of  the missing frequency, thus obtaining an accurate estimate of SDNN24.

Minor comments:

* The close parenthesis is missing in the equation 8.

* Line 171:

... the MESOR (Midline Statistic Of Rhythm, ...

It should be:

... the MESOR (Midline Estimating Statistic Of Rhythm, ...

* Table 5, Abbreviations: Prec, Rec, and F1,  were not defined earlier.

In Table 5 should be:

Precision, Recall,  and F1-score.

Authors’ answer: We would like to thank the reviewer for his useful comments. We edited the manuscript in accordance with his suggestions. In particular, we revise typos throughout the entire manuscript.

Reviewer 2 Report

Bibliography must be improved; the paper only have 19 references with ~20% of auto-cites.

In title and introduction, authors focus in wrist-worn wearable devices, but they only process ECG data from cited databases. This is a trap, because they have not any problem of forearm and hand movements artefacts; that are the big problem of continuous-time PPG on wrist records. New datasets are mandatory, whether from databases or from new data-recording sessions.

There is a possible self-plagiarism between some parts of this article and [1]; e.g., 2.2. Data prepocessing with the same part of [1] - that is not cited by authors.

[1] https://www.mdpi.com/1424-8220/20/24/7122/htm

Author Response

Reviewer comment 1. Bibliography must be improved; the paper only have 19 references with ~20% of auto-cites.

Authors’ answer: Thank you for your comment. We have improved the literature review to provide more details about the background of this study.

Reviewer comment 2. In title and introduction, authors focus on wrist-worn wearable devices, but they only process ECG data from cited databases. This is a trap, because they have not any problem of forearm and hand movements artefacts; that are the big problem of continuous-time PPG on wrist records. New datasets are mandatory, whether from databases or from new data-recording sessions.

Authors’ answer: We agree with the reviewer’s comment. We highlight in the title and throughout the manuscript that we simulate the data from wrist-worn fitness devices by using inter-beats intervals (IBI) obtained from ECG dataset. Moreover, we added a sentence at the end of the discussion session with limitations and future works. As you asserted, IBI data recorded from these devices suffer from high amounts of noise and motion artifacts that negatively affect HRV estimation accuracy. However, wrist-worn devices show reliably results about HR estimation as suggested by several studies both in resting state and during physical activity (e.g., Shcherbina A et al. 2017, Weiler DT et al. 2017, Jo E et al. 2016). As a matter of fact, fitness trackers estimate the HR as their mean in a time window that can be considered a low pass filter on the IBI signal reducing possible estimation error induced by ectopic beats and motion artifacts. Furthermore, due to the fact that HR was found that wrist-worn devices showing a very low error (<5%) we simulated this error by randomly adding a Gaussian errore (0±2 beats per minute) in each HR value created in our datasets (maximum HR tolerance for medical devices defined by CEE). For these reasons, we think that our dataset correctly simulates data derived from these low cost devices. 

References:

  • Shcherbina A et al. Accuracy in Wrist-Worn, Sensor-Based Measurements of Heart Rate and Energy Expenditure in a Diverse Cohort. Journal of Personalized Medicine. 2017, 7(2), 3; https://doi.org/10.3390/jpm7020003
  • Weiler DT et al. Wearable Heart Rate Monitor Technology Accuracy in Research: A Comparative Study Between PPG and ECG Technology. Proceedings of the Human Factors and Ergonomics Society Annual Meeting. 2017, 61(1), 1292-1296; https://doi.org/10.1177/1541931213601804 
  • Jo E et al. Validation of Biofeedback Wearables for Photoplethysmographic Heart Rate Tracking. J Sports Sci Med. 2016, 15(3), 540-547; PMC4974868 

Reviewer comment 3. There is a possible self-plagiarism between some parts of this article and [1]; e.g., 2.2. Data prepocessing with the same part of [1] - that is not cited by authors.

[1] https://www.mdpi.com/1424-8220/20/24/7122/htm

Authors’ answer: Thank you very much for your comment. We revised the entire manuscript in order to avoid any possible self-plagiarism. Moreover, in accordance with your suggestion we have cited our previous work. 

Reviewer 3 Report

1          This paper essentially presents a possible method to extract heart rate variability data from the noisy signals recorded from wrist worn heart rate monitors. They suggest that it may be possible to extract the variability data from the heart rate signal without the need for beat to beat data.

2          The paper is not very easy to read, in part because it uses a very large number of acronyms. There are many places where the English would benefit from editorial attention. For example from line 56  ‘Furthermore, wrist-worn devices normally 57 only report heart rate data (HR), that have proven to be more reliably than NN-interval data when are 58 estimated from these devices [8–11], and seldom report IBI, normally necessary to compute SDNN24.’.

This should read as follows:

Furthermore, wrist-worn devices normally only report heart rate data (HR), as these have been proven to be more reliable than NN-interval data which can only be estimated from these devices, that seldom report the IBI data needed to compute SDNN24.

However, in nearly all cases the sense is still clear.

3          Line 88-91. Why did you choose to simulate the HR signals that might be recorded from a wrist worn device rather than using some real data? Have you any evidence that your simulation is close to what would be recorded from a wrist worn device? You do not appear to include any discussion of the method by which the wrist worn devices recorded heart rate. These are usually optical or impedance devices that are sensitive to blood volume changes and which have a much lower bandwidth that an ECG signal.

4          It seems to me that your validation section (line 176) is really only valid if your simulation of the HR data that might be recorded from a wrist worn device is reasonable. At the very least you need to acknowledge this.

5          Table 2 . Recording numbers to 7 significant figures and the errors to 6 significant figures simply confuses the table.

6          Discussion  line 280

The first sentence reads ’This study shows that SDNN24 can be reliably estimated from HR data, instead of a 24-hours 282 long stream of IBI data.’. I don’t think this has been demonstrated. What does ‘reliably estimated’ mean? You could say ‘This study has shown that SDNN24 data may possibly be estimated form HR data, without the need to have 24 hour IBI data available.’.  

7          Your conclusions are certainly interesting but they obviously need to be validated against actual recordings from wrist worn devices. This needs to be stated.

Author Response

Reviewer comment 1. This paper essentially presents a possible method to extract heart rate variability data from the noisy signals recorded from wrist worn heart rate monitors. They suggest that it may be possible to extract the variability data from the heart rate signal without the need for beat to beat data.

Authors’ answer: We would like to thank the reviewer for his useful comments. We answer point-by-point all of his questions below.

Reviewer comment 2. The paper is not very easy to read, in part because it uses a very large number of acronyms. There are many places where the English would benefit from editorial attention. For example from line 56  ‘Furthermore, wrist-worn devices normally 57 only report heart rate data (HR), that have proven to be more reliably than NN-interval data when are 58 estimated from these devices [8–11], and seldom report IBI, normally necessary to compute SDNN24.’.

This should read as follows:

Furthermore, wrist-worn devices normally only report heart rate data (HR), as these have been proven to be more reliable than NN-interval data which can only be estimated from these devices, that seldom report the IBI data needed to compute SDNN24.

However, in nearly all cases the sense is still clear.

Authors’ answer: We have reduced the number of acronyms in order to improve the manuscript’s readability. Moreover, the text was revised by English mothertough. 

Reviewer comment 3. Line 88-91. Why did you choose to simulate the HR signals that might be recorded from a wrist worn device rather than using some real data? Have you any evidence that your simulation is close to what would be recorded from a wrist worn device? You do not appear to include any discussion of the method by which the wrist worn devices recorded heart rate. These are usually optical or impedance devices that are sensitive to blood volume changes and which have a much lower bandwidth that an ECG signal.

Authors’ answer: This study is focused to simulate data derived from wrist-worn devices equipped with HR sensors in order to assess if it is possible to accurately estimate SDNN24 from data recorded by these low cost devices. Differently to HRV, HR data derived from wrist-worn devices is found to be reliable from several previous studies both in resting state and during physical activity (e.g., Shcherbina A et al. 2017, Weiler DT et al. 2017, Jo E et al. 2016). Fitness trackers estimate the HR as the mean of inter-beats intervals (IBI) in a time window that can be considered a low pass filter on IBI signal that reduces any possible estimation error induced by ectopic beats and motion artifacts. For these reasons, we think that our dataset accurately simulates data derived from these low cost devices. We hence better explain these points throughout the manuscript.

We hence highlight in the title and throughout the manuscript that we simulate the data from wrist-worn fitness devices by using inter-beats intervals (IBI) obtained from ECG dataset.

References:

  • Shcherbina A et al. Accuracy in Wrist-Worn, Sensor-Based Measurements of Heart Rate and Energy Expenditure in a Diverse Cohort. Journal of Personalized Medicine. 2017, 7(2), 3; https://doi.org/10.3390/jpm7020003
  • Weiler DT et al. Wearable Heart Rate Monitor Technology Accuracy in Research: A Comparative Study Between PPG and ECG Technology. Proceedings of the Human Factors and Ergonomics Society Annual Meeting. 2017, 61(1), 1292-1296; https://doi.org/10.1177/1541931213601804 
  • Jo E et al. Validation of Biofeedback Wearables for Photoplethysmographic Heart Rate Tracking. J Sports Sci Med. 2016, 15(3), 540-547; PMC4974868 

Reviewer comment 4. It seems to me that your validation section (line 176) is really only valid if your simulation of the HR data that might be recorded from a wrist worn device is reasonable. At the very least you need to acknowledge this.

Authors’ answer: As explained in your previous comments, HR data is accurately estimated by wrist-worn devices. For this reason, our results reflect real estimation of SDNN24 by HR data derived from these low cost devices. However, future works are required to estimate any possible influence of the missing time window on SDNN24 estimation by using our approach. We added a sentence at the end of the discussion session with limitations and future works.

Reviewer comment 5. Table 2 . Recording numbers to 7 significant figures and the errors to 6 significant figures simply confuses the table.

Authors’ answer: Thank you for your suggestion. We have edited the table by using scientific notation numbers.

Reviewer comment 6. Discussion  line 280

The first sentence reads ’This study shows that SDNN24 can be reliably estimated from HR data, instead of a 24-hours 282 long stream of IBI data.’. I don’t think this has been demonstrated. What does ‘reliably estimated’ mean? You could say ‘This study has shown that SDNN24 data may possibly be estimated form HR data, without the need to have 24 hour IBI data available.’.  

Authors’ answer: Thank you for your useful comments. We edited this sentence in accordance with your suggestion.

Reviewer comment 7. Your conclusions are certainly interesting but they obviously need to be validated against actual recordings from wrist worn devices. This needs to be stated.

Authors’ answer: we agree with you. We added a sentence at the end of the discussion session with limitations and future works: “It should be noted that one of the most important limitation of this study is that our approach is validated on simulated HR data derived from wrist-worn fitness tracker by using ECG data that should not be affected by motion artefacts. However, in order to better simulate data from these low cost devices, we introduce an error of about 0±2 beats per minute in each time window HR value that is the maximum HR tolerance for medical devices accepted by CEE. However, future works are needed in order to validate our approach on HR data recordings from wrist worn devices.”

Round 2

Reviewer 2 Report

Bibliography must be improved again; now, the paper has 29 references with 20.7% of self-cites (6).

In title and introduction, authors focus on wrist-worn wearable devices, however the ECG data that they use is not collected using said devices, but is from an already existing database. This time around, the authors included a paragraph advising that the data come from ECG, in section 2.4.

Nevertheless, this is still deceiving and confusing for future readers. For instance, authors that may conduct a  scooping review, would select an interesting article through title and abstract. The title suggests that the article is about wrist-wron wearable, when it is not the case.

This could be easily solved by changing the focus of the title, abstract and introduction. In the Discussion section, the authors could suggest that those analysis are comparable to PPG obtained from a wearable.

Aditionally, the authors do not simulate PPG data from ECG, which is something that the reader would expect to be done in the article - me first. 

I recommend removing the confusing references to wrist-worn wearable in the above cited sections, and that the authors simply defend their "high"-noise-IBI procedure.

Author Response

Reviewer comments

Bibliography must be improved again; now, the paper has 29 references with 20.7% of self-cites (6).

In title and introduction, authors focus on wrist-worn wearable devices, however the ECG data that they use is not collected using said devices, but is from an already existing database. This time around, the authors included a paragraph advising that the data come from ECG, in section 2.4.

Nevertheless, this is still deceiving and confusing for future readers. For instance, authors that may conduct a  scooping review, would select an interesting article through title and abstract. The title suggests that the article is about wrist-wron wearable, when it is not the case.

This could be easily solved by changing the focus of the title, abstract and introduction. In the Discussion section, the authors could suggest that those analysis are comparable to PPG obtained from a wearable.

Aditionally, the authors do not simulate PPG data from ECG, which is something that the reader would expect to be done in the article - me first. 

I recommend removing the confusing references to wrist-worn wearable in the above cited sections, and that the authors simply defend their "high"-noise-IBI procedure.

Authors’ answer

We would like to thank the reviewer for his/her useful comments. We agree with all of these comments and we edited the paper after his/her suggestions, in our opinion strengthening and greatly improving our paper. We have added a subsection in Methods where we study the error of estimating HR from wrist worn PPG devices using ECG as ground truth. As the reviewer asserted in his/her revision, we use data derived from ECG and not from PPG. These datasets are not affected by motion artefacts, but, to replicate HR data from low-cost wrist-worn devices, we simulated the PPG error adding a random gaussian error to average HR from different time window length. The wrist-worn device provides the average of HR in a time window length and not the instant HR values because it could be affected by motion artefacts. In order to assess this error, we compare the average of the instant HR in different time window length (i.e., 1, 5, 10, 30 and 60 minutes) between ECG (Polar H7 heart rate monitor - Polar Electro Inc., Bethpage, NY, USA) and PPG (BioBeam - BioBeats Group Ltd, London, www.biobeats.com) data from MMASH dataset. Unfortunately, for some technical issues during data recording, we have  HR data for both the devices only for a small, but still significant, part of the day (only a few hours per participants, not enough to be used as the main dataset for the paper, but enough to characterise PPG sensor noise vs ECG ground truth). We found a bias between the two devices that vary between 0.03±5.91 bpm (1 minute time window) and -0.35±3.95 bpm (60 minutes time window). Thanks to this analysis we can randomly add a Gaussian error to each time window in accordance with the bias estimated in each time window in order to better simulate HR data derived from PPG devices. For this reason, we are sure that we accurately simulated PPG data from ECG. The introduction of this random Gaussian error does not affect the estimation of SDNN24 showing similar results compared to our previous results obtained when we simulate the HR estimation error in accordance with CEE guidelines for medical devices (0±2 bpm). This can be explained by the fact that motion artifact seems to add high frequency noise, without affecting lower frequencies; as explained in the paper low frequencies are the main contributors to SDNN24, and our model is designed to capture mainly circadian effects, disregarding higher frequencies, therefore resilient to noise like motion artifacts.

For this reason, we decided to not change the focus of our paper due to the fact that the main rationale to make this study is to find a way to estimate SDNN24 by HR data derived from wrist-worn devices. As a matter of fact, this approach permits to estimate SDNN24 from data derived from low-cost wearable devices that nowadays are not possible due to the fact that inter-beats data are higher affected by motion artefacts. However, we edited the title as “SDNN24 estimation from semi-continuous HR measures” to avoid too much focus on wearable data simulation but highlighting the fact that average HR data in different time window lengths is enough to estimate SDNN24.

Moreover, in accordance with the Reviewer suggestions, we improve the literature review for increasing the number of relevant references. However, we could not reduce the number of self-citations due to the fact that all these citations are highly relevant and necessary to explain and support our approach:

  • Dataset description (required from Physionet):
    • Rossi A et al. Multilevel Monitoring of Activity and Sleep in Healthy People (version 1.0.0). Physionet. 2020.
    • Rossi, A.; Da Pozzo, E.; Menicagli, D.; Tremolanti, C.; Priami, C.; Sirbu, A.; Clifton, D.; Martini, C.; Morelli, D. A Public Dataset of 24-h Multi-Levels Psycho-Physiological Responses in Young Healthy Adults. Data 2020,5. doi:10.3390/data5040091.
  •  
  • Motion artefacts reflect on HRV estimation:
    • Morelli, D.; Rossi, A.; Cairo, M.; Clifton, D. Analysis of the Impact of Interpolation Methods of Missing RR-Intervals Caused by Motion Artifacts on HRV Features Estimations. Sensors 2019,19, 3163.doi:10.3390/s19143163.15.
    • Morelli, D.; Bartoloni, L.; Colombo, M.; Clifton, D. Profiling the propagation of error from PPG to HRV features in a wearable physiological-monitoring device. Healthcare technology letters 2018,5, 59–64.
    • Rossi, A.; Pedreschi, D.; Clifton, D.; Morelli, D. Error Estimation of Ultra-Short Heart Rate Variability Parameters: Effect of Missing Data Caused by Motion Artifacts. Sensors 2020,20. doi:10.3390/s20247122.
  • Multiple component circadian rhythm:
    • Morelli, D.; Bartoloni, L.; Rossi, A.; DA, C. A computationally efficient algorithm to obtain an accurate and interpretable model of the effect of circadian rhythm on resting heart rate. Physiological Measurements 2019, 40. doi:10.1088/1361-6579/ab3dea.